# Comparative proteomics analysis reveals the molecular mechanism of enhanced cold tolerance through ROS scavenging in winter rapeseed (*Brassica napus* L.)

**Wenbo Mi****, Zigang Liu\*, Jiaojiao Jin, Xiaoyun Dong, Chunmei Xu, Ya Zou, Mingxia Xu, Guoqiang Zheng, Xiaodong Cao, Xinling Fang, Caixia Zhao, Chao Mi**

Gansu Provincial Key Laboratory of Aridland Crop Science, College of Agronomy, Gansu Agricultural University, Lanzhou, China

\* lzgworking@163.com

**Data Availability Statement:** All relevant data are within the paper and its Supporting Information files.

## Abstract

Two winter rapeseed cultivars, "NS" (cold tolerant) and "NF" (cold sensitive), were used to reveal the morphological, physiological, and proteomic characteristics in leaves of plants after treatment at -4˚C for 12 h(T1) and 24 h(T2), and at room temperature(T0), to understand the molecular mechanisms of cold tolerance. Antioxidant activity and osmotic adjustment ability were higher, and plasma membrane injury was less obvious, in NS than in NF under cold stress. We detected different abundant proteins (DAPs) related to cold tolerance in winter rapeseed through data-independent acquisition (DIA). Compared with NF, A total of 1,235 and 1,543 DAPs were identified in the NSs under T1 and T2, respectively. Compared with NF, 911 proteins were more abundant in NS only after cold treatment. Some of these proteins were related to ROS scavenging through four metabolic pathways: lysine degradation; phenylalanine, tyrosine, and tryptophan; flavonoid biosynthesis; and ubiquinone and other terpenoid-quinone biosynthesis. Analysis of these proteins in the four candidate pathways revealed that they were rapidly accumulated to quickly enhance ROS scavenging and improve the cold tolerance of NS. These proteins were noticeably more abundant during the early stage of cold stress, which was critical for avoiding ROS damage.

## Introduction

Cold temperatures have adverse effects on the growth and development of plants [1]. Winter rapeseed (*Brassica napus*) is one of the few crop species able to survive frosty winters in temperate regions [2]. However, only strong cold-tolerant cultivars can survive frigid winters, while cultivars with only slight cold tolerance often wither during winters [3]. Plants originating from temperate regions frequently encounter low temperatures in the early spring and late autumn; furthermore, perennial or overwintering crops, such as holly [4], winter wheat [5] and winter rape [2], normally experience cold stress while overwintering. Plants have evolved several mechanisms to adapt to low temperatures, including cold-adapted regulation in physiological, biochemical, and molecular processes [6].

**Funding:** This work was supported by the National Natural Science Foundation of China (31660404), National Key Basic Research and Development Program (2018YFD0100500), University of Gansu Province Scientific Research Achievement Transformation and Cultivation Project (2018D-13), Special Fund for the Construction of Modern Agricultural Industrial Technology System of Gansu Province (17ZD2NA016-4); Special funds for the central government to guide local technological development. The funders had no role in study design, data collection and analysis, decision to publish, or preparation of the manuscript.

**Competing interests:** The authors have declared that no competing interests exist.

**Abbreviations:** DAPs, Different abundant proteins; DIA, Data Independent Acquisition; ROS, Reactive oxygen species; SOD, Superoxide dismutase; CAT, Catalase; APX, Ascorbate peroxidase; MDA, Malondialdehyde; POD, Peroxidase; SS, Soluble sugar; SP, Soluble protein; ASC, Ascorbic acid; GSH, Glutathione; Itraq, The isobaric tags for relative and absolute quantification; MAS, Marker-assisted selection; GO, Gene ontology; KEGG, Kyoto Encyclopedia of Genes and Genomes; qRT-PCR, Quantitative real-time polymerase chain reaction; FLS, Flavonol synthase; VTE1, Naphthoate synthase; CA4H, Trans-cinnamate4-monooxygenase; 4CL, 4-coumarate—CoA ligase; GB, Glycine betaine; AtoB, Acetyl-CoA C-acetyltransferase; ALDH, Aldehyde dehydrogenase; TrpD, Anthranilate phosphoribosyltransferase; TAT, Tyrosine aminotransferase; GOT1, Aspartate aminotransferase, cytoplasmic.

Reactive oxygen species (ROS) accumulate in plants under cold stress [7]. For example, levels of hydrogen peroxide were more than triple in winter rapeseed under cold stress for 72 h than plants at room temperature [2]. ROS content increased in both *indica* and *japonica* rice under cold stress, but the increase was lower in japonica rice compared with *indica* rice; in addition, significant differences in ROS-related metabolites were detected between these two types of rice [8, 9]. ROS have the janus function of inducing adaptability or producing peroxidative injury to organisms, which primarily depends on the levels of ROS in plants. High levels of ROS are toxic to cells, and low levels permit small increases that can trigger signaling pathways that induce responses among downstream genes and enhance adaptability under biotic and abiotic stress [10].

To assume their signaling roles, ROS must reach certain threshold levels to trigger downstream responsive factors in the signal pathway, while also preventing ROS content from passing the threshold that would cause oxidative damage [11]. Therefore, ROS levels must be tightly regulated by antioxidants: both antioxidase proteins (such as superoxide dismutase, SOD; peroxidase, POD; and catalase, CAT) and non-enzymatic substances (such as quercetin, flavonoids, tryptophan, betaine, and glutathione) contribute to metabolic pathways promoted by enzymatic proteins [10]. In winter rapeseed, the activity of SOD, POD, CAT, and APX was significantly increased to stabilize ROS at an appropriate level under cold stress [2]. ROS permit small increases that can induce the accumulation of downstream cold-responsive proteins and osmotic substances through the MKK6-MAPK3 cascade reaction in plants, thereby enhancing tolerance to cold stress [12, 13].

Proteins can directly affect phenotypes by controlling a series of molecular processes in plants, including signaling transduction pathways, the genic expression process, and secondary metabolic processes [14]. Proteins perform several biological functions, and their abundance depends on transcription and post-transcriptional translation [12]. Nevertheless, transcriptomics analysis has been used to identify different transcriptional genes in winter oilseed rape (*Brassica napus*) treated at various cold temperatures. Although genes might be transcribed, their transcription does not necessarily indicate that their corresponding proteins are present [15]. In other words, a high transcriptional level does not necessarily imply the existence of a biological phenotype. Therefore, comparative analysis at the proteomics level is essential for furthering our understanding of cold-tolerance mechanisms in plants.

High-throughput protein sequencing technology, including iTRAQ (isobaric tags for relative and absolute quantification) and DIA (data-independent acquisition), have been widely used for the quantitative comparative analysis of proteomes and the identification of different abundant proteins (DAPs) of plants under various cold temperatures [16, 17]. In jojoba, quantitative proteomic analysis has shown that differentially accumulated proteins are closely associated with cold stress through various molecular processes, such as lipid metabolism and transport, adjustment of the cytoskeleton, and ROS scavenging. In addition, cold-regulated 47 factor and three other candidate proteins might play important roles in cold stress [6]. The molecular basis of cold-tolerance has been assessed by comparing proteomic divergence between two cultivars of winter turnip rape [14]. Furthermore, plant proteomes of many species have been analyzed to reveal cold-tolerance mechanisms [18–21].

Under extremely low temperatures, winter turnip rape (*Brassica rapa*) is the only oil crop that can survive frosts during the winter in the northern region of China. However, winter oilseed rape was not able to overwinter in this region until strong cold-tolerant cultivars were bred by our research group from Gansu Agricultural University. "17NTS57" is currently the most cold-tolerant variety of oilseed rape, as it can survive temperatures as low as -26˚C and has an overwintering rate above 90% without snow cover. Comparative proteomic analysis of the strong cold-tolerant variety "NS" is necessary for furthering our understanding of the

molecular mechanisms of cold tolerance, to identify key cold-responsive genes, and to provide a knowledge base that could be used to breed and improve the cold tolerance of winter rapeseed.

## Results

### Morphological and physiological responses to cold stress

Seedlings with 5~6 leaves of two cultivars (NS and NF) were cold treated at 4°C for 12 h (T1) and 24 h (T2) to study morphological and physiological responses to cold stress. Leaves are the most sensitive component of plants to cold stress as a consequence of their exposed position. Slight injuries were only present on the lower leaves of NS, but whole NF plants experienced severe dehydration under cold stress (Fig 1A). Activity of SOD, POD, and CAT was used to assess ROS-scavenging ability, soluble sugar, and soluble protein as osmotic regulative substances, while MDA content and conductivity were used to assess the degree of peroxidation in plants under cold stress. The activity of SOD, POD, CAT, and the content of soluble sugar and soluble protein was higher in NS than in NF under cold stress. In contrast, MDA content and relative conductivity were lower in NS than in NF under cold treatment for 12 h and 24 h. Specifically, MDA content in NF was more than two times greater than that in NS, and the relative conductivity of NF was 1.66 times greater than that of NS under cold stress for 12 h. These results suggest that more damage was present in NF after cold treatment for 12 h. This finding was also verified through observations of seedling morphology (Fig 1B).

### Identification of DAPs using DIA technology

The DAPs between six samples from two varieties (NS and NF) at 4°C for 12 h (T1) and 24 h (T2) and room temperature (T0) as the control were identified and quantified using DIA and LC-MS/MS technology. Consequently, a total of 38804 precursors were produced, and 31042 unique peptides and 14002 proteins were identified with a false discovery rate less than 1% (S1 Table). Among these proteins, a total of 1978 DAPs in NST0_NFT0, 1235 DAPs in NST1_NFT1, and 1543 DAPs in NST2_NFT2 were identified based on the criteria of $p < 0.05$ for the adjusted p-value and greater than 1 for the log2 base of fold change (S2 Table). The bar charts reflect the number of more- and less-abundant DAPs between the two varieties under cold stress and in the control treatment. Under room temperature, 896 DAPs were more abundant and 1082 DAPs were less abundant in NS than in NF. After 12 h of cold stress, 724 DAPs were more abundant, and 511 DAPs were less abundant, in NS than in NF. Similarly, 452 DAPs were more abundant, and 1091 DAPs were less abundant, in NS than in NF after 24 h of

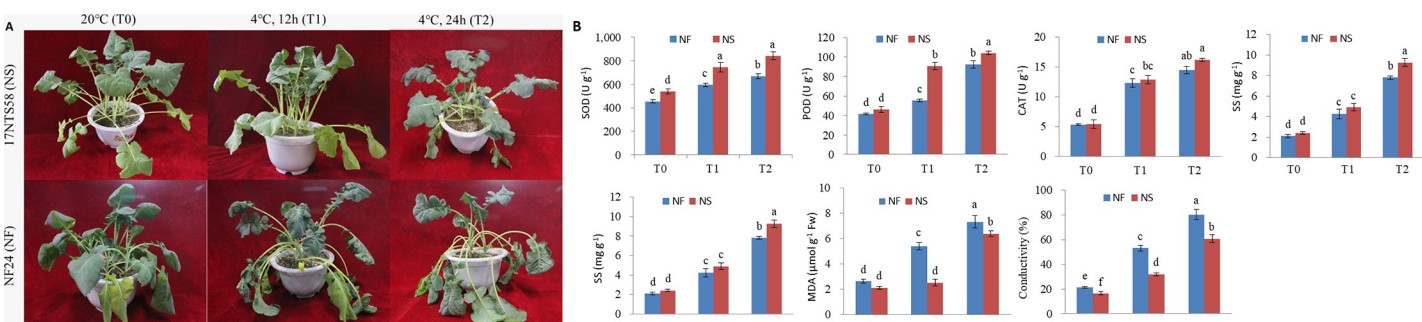

**Fig 1. Morphological characteristics and physiological indexes of winter rapeseed seedling under cold stress and room temparature condition.** (A) Morphological characteristics of NS and NF; (B) Physiological indexes of NS and NF under cold stress and room temparature. Error bars denote standard error of the mean. Significant differences between various treatments at p ≤ 0.05 are denoted by a various lowercase.

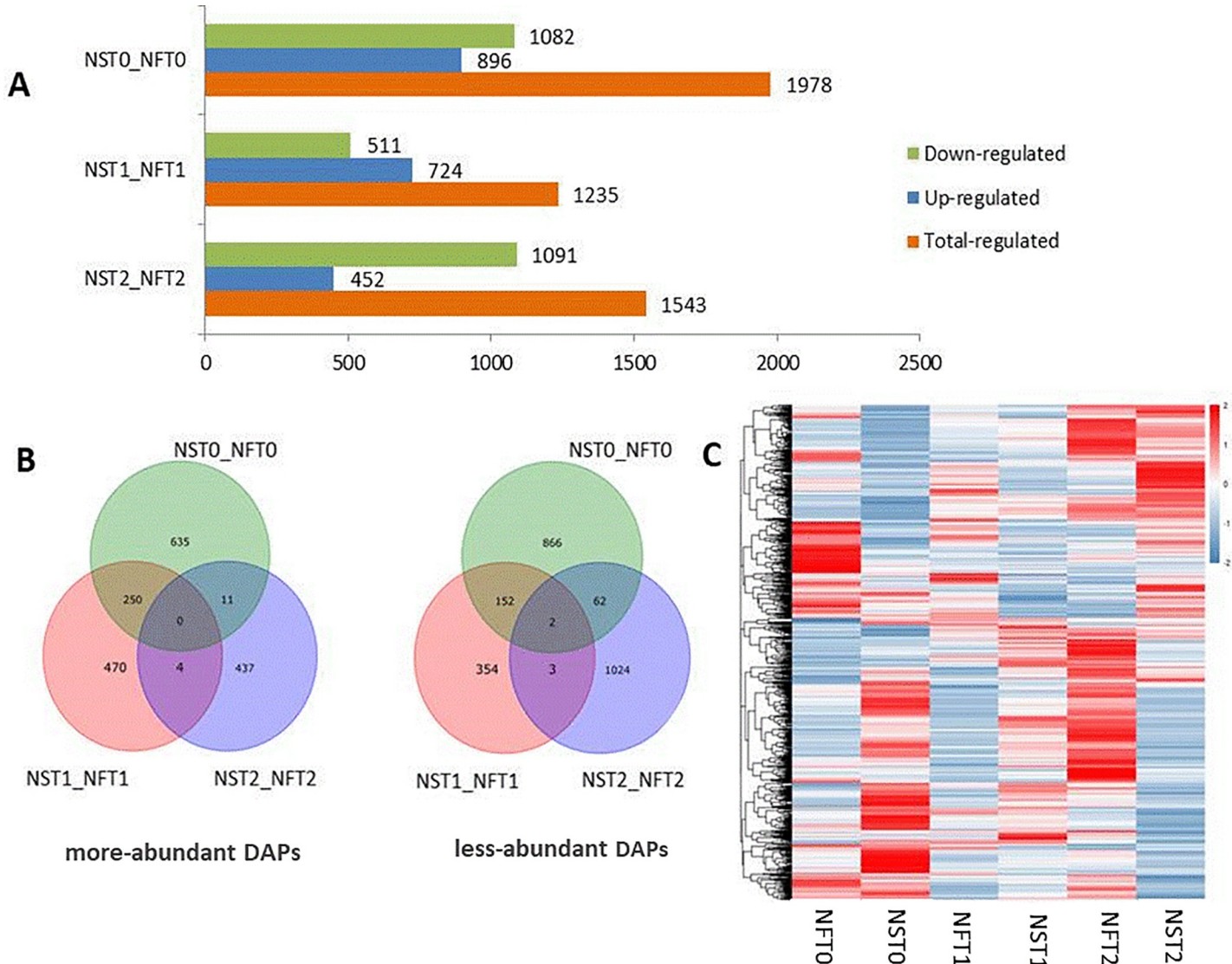

**Fig 2. DAPs between two varieties at three cold-treatments of winter rapeseed (*Brassica napus*).** (A) Column diagram of the DAPs. (B) Venn diagrams of the DAPs between two varieties, showing the numbers of the overlaps or unique DAPs (left: more-abundance, right: less-abundance). (C) The heat maps showing DAPs accumulation in each group, color change from red to blue represented abundant change from more to little.

cold stress. In addition, these results indicated that more proteins, along with the added accumulation of cold stress during early stages, were important for enhancing the cold tolerance of NS when the environmental temperature suddenly decreased (Fig 2A).

By comparing all of the DAPs between the two varieties at three treated time points, the Venn diagram shows the number of more- or less-abundant DAPs as well as specific and commonly abundant DAPs in the different groups(Fig 2B). A total of 1501 DAPs (635 more abundant and 866 less abundant) in NST0_NFT0, 824 DAPs (470 more abundant and 354 less abundant) in NST1_NFT1, and 1461 DAPs (437 more abundant and 1024 less abundant) in NST2_NFT2 were identified. Two common DAPs that were less abundant in NS were present in all treatments, while three DAPs were less accumulated in NS only under cold stress. A total of 470 more-abundant DAPs in NS were exclusively identified at 12 h of cold stress, while 437 more-abundant DAPs in NS were uniquely found at 24 h of cold stress. Four common DAPs

that were more abundant in NS were identified at 12 h and 24 h of cold treatment but not under treatment at room temperature. These unique proteins, which showed more accumulation in NS only under cold conditions, appear to play an important role in improving the cold tolerance of winter rapeseed.

## GO and KEGG pathway annotations for DAPs

To identify the different DAPs that play a role in the cold responses of the two varieties, we used GO analysis to functionally classify DEGs. These DAPs were divided into three categories of GO: biological process, molecular function, and cellular component (Fig 3; S3 Table). The significant enrichment terms were used to screen DAPs between the two varieties at three cold-treated time points. In NST0_NFT0, higher GO terms were associated with rhythmic process, metabolic process, growth, and multiply, which reflected the characteristic divergence between the two varieties (Fig 3A). In NST1_NFT1, 1235 DAPs were significantly enriched in 46 terms (Fig 3B), while 1543 DAPs were significantly enriched in 45 terms in NST2_NFT2 (Fig 3C). Some of these terms were related to the cold response, such as response to stimulus, signaling and signal transducer activity, antioxidant activity, biological regulation, transcription factors, and metabolic processes.

To annotate the functions of proteins, we conducted a pathway enrichment analysis of the DAPs in NST0_NFT0, NST1_NFT1, and NST2_NFT2 based on the KEGG database (Fig 4).

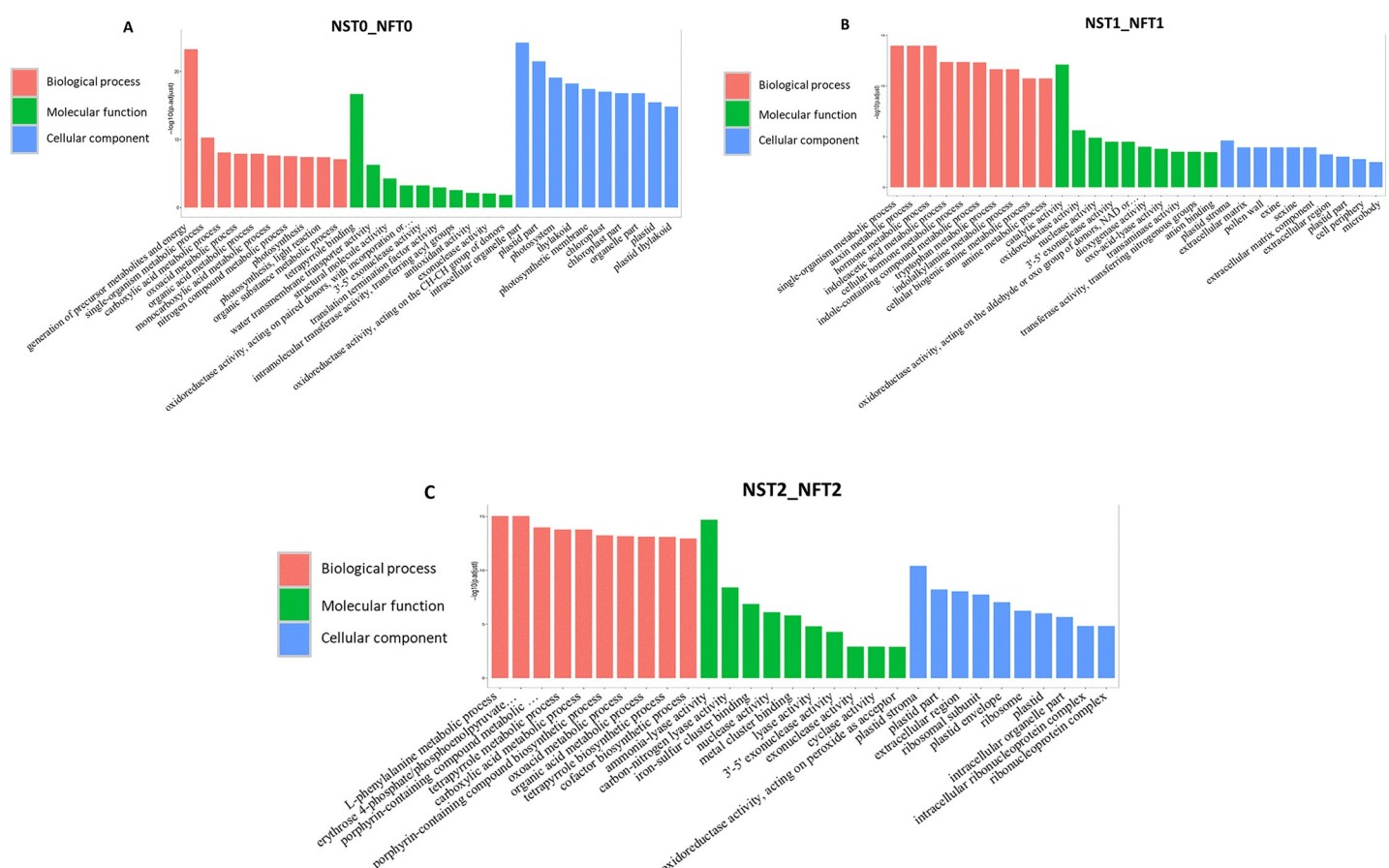

**Fig 3.** GO enrichment analysis of DAPs between two varieties under cold stress for 12 h (B) and 24 h (C), and control treatment (A).

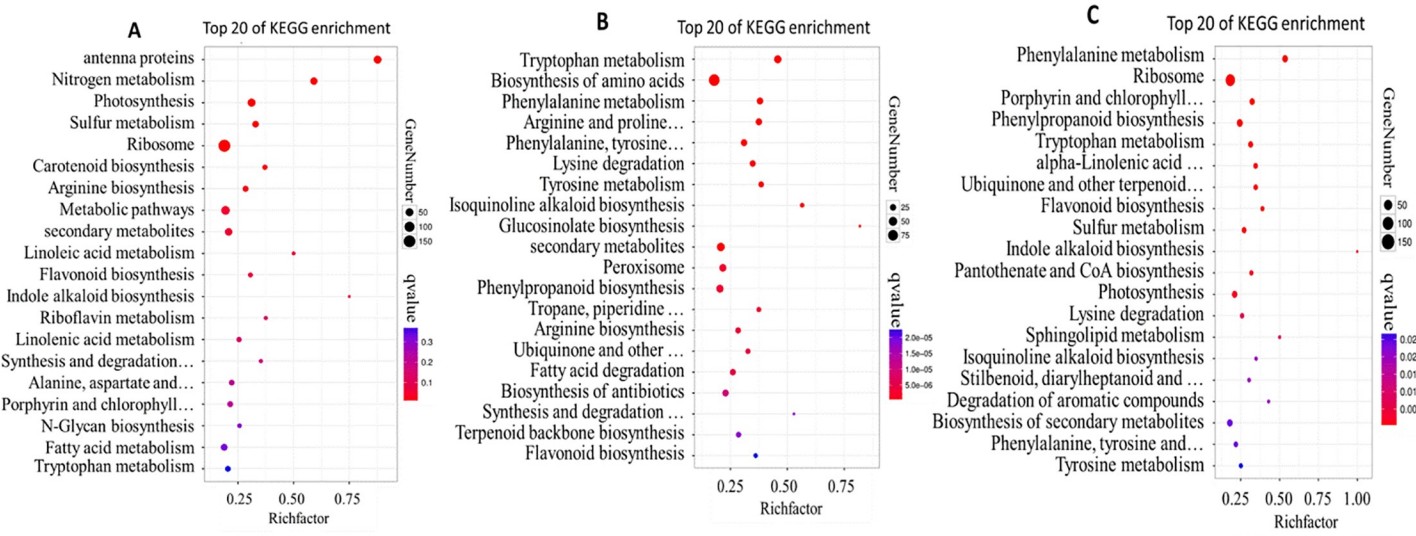

**Fig 4.** KO pathways enrichment analysis of DAPs between two varieties under cold for 12 h (B) and 24 h (C), and control treatment (A).

In NST0_NFT0, 885 DAPs were significantly enriched in 7 pathways (corrected $p < 0.05$ and $Q < 0.05$) (Fig 4A). In NST1_NFT1, 583 DAPs were significantly enriched in 42 pathways (Fig 4B). In NST2_NFT2, 721 DAPs were significantly enriched in 30 pathways (Fig 4C). An additional comparative analysis indicated that four pathways—ko0400: phenylalanine, tyrosine and tryptophan biosynthesis; ko00130: ubiquinone and other terpenoid-quinone biosynthesis; ko00941: flavonoid biosynthesis; and ko00310: lysine degradation—were common between NST1_NFT1 and NST2_NFT2, and these four pathways were not detected in NST0_NFT0. Further analysis showed that these four pathways were related to the cold response through the production of ROS scavengers. Thus, these four pathways were selected as candidate pathways for additional study.

### Identification of DAPs correlated to candidate pathways

A total of 80 and 68 DAPs that were detected in NST1_NFT1 and NST2_NFT2, respectively, were associated with the four candidate pathways. There were 44 common DAPs in T1 and T2, including 34 more-abundant DAPs that were involved in phenylalanine, tyrosine and tryptophan biosynthesis; phenylalanine metabolism; flavonoid biosynthesis; and lysine degradation (Fig 5A; Table 1; S4 Table). These DAPs likely affected the cold resistance of winter rapeseed. After functionally annotating these proteins, we found that the DAPs represented 18 crucial enzymes (Fig 5B).

### Validation of DAPs and their corresponding genes by qRT-PCR

To further verify the relationship between protein accumulation and the degree of transcription of corresponding genes, 23 DAPs were selected in the candidate pathways using high-throughput sequencing to detect the mRNA levels of their corresponding genes using quantitative real-time PCR (qRT-PCR) (Fig 6; S5 and S6 Tables). The trends in expression of 21 selected genes were similar to the trends in the accumulation of their corresponding proteins. Two of the selected genes showed the opposite transcriptional pattern to patterns of accumulation of their corresponding proteins, which might be explained by differences in the fate of

**A**                                    **B**

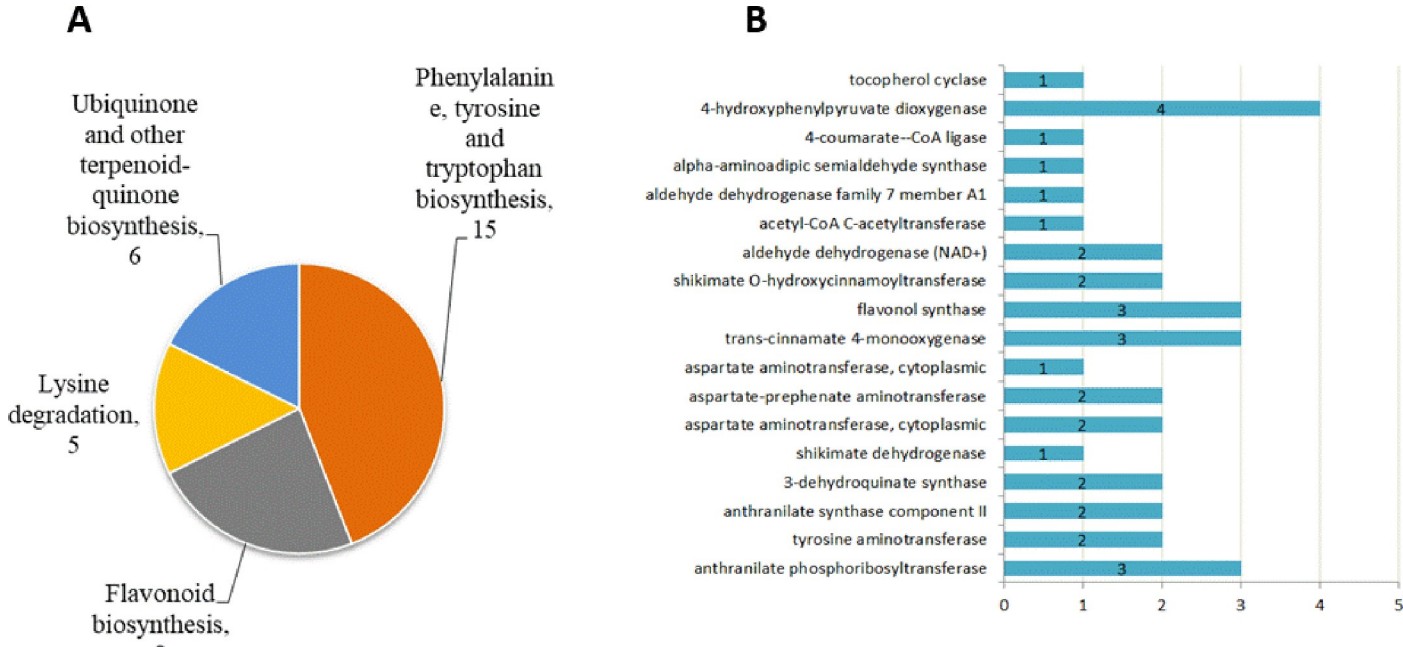

**Fig 5. KO and GO annotation of DAPs in the candidate pathway correlated with cold tolerance of winter rapeseed.** A: the candidate pathway and DAP numbers in each candidate pathway. B: GO annotation of DAPs, numbers in histogram indicate number of DAPs.

mRNA as a consequence of post-transcriptional regulation, such as mRNA turnover, translation rate, and translation.

In conclusion, we constructed a model describing how this candidate proteins are involved in cold resistance by mediating the ROS in plants (Fig 7). In this study, the abundance of FLS, trpD and ALDH proteins was significantly higher in NS under cold stress; FLS is a synthase catalyzing the biosynthesis of quercetin. Quercetin is one of the most important products of the flavonoid biosynthesis pathway and is a highly active non-enzymatic scavenger of ROS. TrpD can catalyze the conversion of anthranilate to tryptophan, and tryptophan can enhance the activity of SOD and GSH-PX as well as suppress the accumulation of MDA. The formation of butyro-betaine, the precursor of betaine, via ALDH catalysis is necessary for ensuring a sufficient supply of betaine, which is a non-enzymatic ROS scavenger that participates in the cold-tolerant pathway of winter rapeseed.

## Discussion

Winter rapeseed is not only an important oilseed crop but also an important winter cover crop. Wind erosion on farmland is severe because of the dry climate in northern China, and the cover that it provides to farmland can generate major ecological benefits by suppressing wind erosion. However, low overwinter survival rate due to low temperatures and frost is one of the major challenges of cultivating winter rapeseed in this region. Improving the cold tolerance of cultivars is critical to the successful overwintering of winter rapeseed. Therefore, there is an urgent need to study the cold tolerance mechanisms of winter oilseed rape [2].

Plants, including winter rapeseed, exposed to cold conditions during the early spring and late autumn, or extremely low temperatures generally during the winter, are damaged as evidenced by the presence of ROS accumulation, membrane rigidification, protein destabilization, metabolic disequilibrium, and the formation of ice crystals in the extracellular and

**Table 1. Candidate proteins from the four pathways associated with cold tolerance of winter rapeseed.**

| Pathway | Common DAP | KO id | Annotation [EC no.] | Fold (NST1/NFT1) | Regulation | Fold (NST2/NFT2) | Regulation | q-value |
|---|---|---|---|---|---|---|---|---|
| Phenylalanine, tyrosine and tryptophan biosynthesis | XP_013728819.1 | K00766 | anthranilate phosphoribosyltransferase (trpD)[EC:2.4.2.18] | 2.46 | Up | 0.41 | Down | 0.00 |
| | XP_013676058.1 | | | 2.46 | Up | 0.41 | Down | 0.00 |
| | XP_013722305.1 | | | 1.74 | Up | 0.47 | Down | 0.00 |
| | XP_013653143.1 | K00815 | tyrosine aminotransferase(TAT)[EC:2.6.1.5] | 1.87 | Up | 0.47 | Down | 0.00 |
| | XP_013718039.2 | | | 1.74 | Up | 0.47 | Down | 0.00 |
| | XP_013657201.1 | K01658 | anthranilate synthase component II(trpG) [EC:4.1.3.27] | 1.62 | Up | — | — | 0.00 |
| | XP_022545953.1 | | | 1.62 | Up | — | — | 0.00 |
| | XP_022565834.1 | K01735 | 3-dehydroquinate synthase(aroB) [EC:4.2.3.4] | 1.52 | Up | — | — | 0.00 |
| | XP_013661385.1 | | | 1.52 | Up | — | — | 0.00 |
| | XP_013640777.1 | K13832 | 3-dehydroquinate dehydratase(aroDE) [EC:1.1.1.25] | 2.30 | Up | 0.54 | Down | 0.00 |
| | XP_013726187.1 | K14454 | aspartate aminotransferase, cytoplasmic (GOT1)[EC:2.1.6.6] | 0.50 | Down | 2.30 | Up | 0.00 |
| | XP_013721995.1 | | | 11.31 | Up | — | — | 0.00 |
| | XP_013663182.1 | | | 2.30 | Up | 0.47 | Down | 0.00 |
| | XP_013664180.1 | K15849 | bifunctional aspartate aminotransferase and glutamate(PAT)[EC:2.6.1.78] | 2.00 | Up | — | — | 0.00 |
| | XP_013707917.2 | | | 2.00 | Up | — | — | 0.00 |
| Flavonoid biosynthesis | XP_022571603.1 | K00487 | trans-cinnamate 4-monooxygenase (CYP73A)[EC:1.14.13.11] | 2.00 | Up | 0.38 | Down | 0.00 |
| | XP_022557738.1 | | | 2.00 | Up | 0.38 | Down | 0.00 |
| | XP_013683255.1 | | | 2.00 | Up | 0.38 | Down | 0.00 |
| | XP_013678442.1 | K05278 | flavonol synthase(FLS) [EC:1.14.11.23] | 1.74 | Up | 0.47 | Down | 0.00 |
| | XP_022562120.1 | | | 1.74 | Up | 0.47 | Down | 0.00 |
| | XP_013715671.1 | | | 2.83 | Up | — | — | 0.00 |
| | XP_013743978.1 | K13065 | shikimate O-hydroxycinnamoyltransferase (HCT) [EC:2.3.1.133] | 2.64 | Up | 0.23 | Down | 0.00 |
| | XP_013692583.1 | | | 2.64 | Up | 0.23 | Down | 0.00 |
| Lysine degradation | XP_013672989.1 | K14085 | aldehyde dehydrogenase family 7 member A1(ALDH7A1)[EC:1.2.1.31] | 0.33 | Down | 3.48 | Up | 0.00 |
| | XP_013707249.1 | K00128 | aldehyde dehydrogenase (NAD+)(ALDH) [EC:1.2.1.3] | 2.46 | Up | — | — | 0.00 |
| | XP_013745666.1 | | | 2.64 | Up | — | — | 0.00 |
| | XP_022565853.1 | K00626 | acetyl-CoA C-acetyltransferase(atoB) [EC:2.3.1.9] | 0.38 | Down | 1.87 | Up | 0.00 |
| | XP_022550217.1 | K14157 | alpha-aminoadipic semialdehyde synthase (AASS)[EC:1.5.1.8] | 0.38 | Down | 3.25 | Up | 0.00 |
| Ubiquinone and other terpenoid-quinone biosynthesis | XP_013749046.1 | K01904 | 4-coumarate—CoA ligase(4CL)[EC:6.2.1.12] | 1.62 | Up | — | — | 0.00 |
| | XP_013723238.1 | K00457 | 4-hydroxyphenylpyruvate dioxygenase (HPD, hppD)[EC:1.13.11.27] | 0.35 | Down | 3.73 | Up | 0.00 |
| | XP_013695640.1 | | | 0.35 | Down | 3.73 | Up | 0.00 |
| | XP_013723237.1 | | | 0.35 | Down | 3.73 | Up | 0.00 |
| | XP_013695641.1 | | | 0.35 | Down | 3.73 | Up | 0.00 |
| | XP_013719625.1 | K09834 | tocopherol cyclase(VTE1, SXD1) [EC:5.5.1.24] | — | — | 1.62 | Up | 0.00 |

"-" means no significant differential-accumulation.

intracellular space [6]. Major crop species that originated from temperate regions, including winter wheat, rye, and winter rapeseed, have evolved complex mechanisms to adapt to and survive in low-temperature conditions (Shi et al. 2018). ROS play vital roles in mechanisms underlying the cold response and in early cold-signaling perception and transduction [7]. ROS produced in plants have janus functions [10], primarily by causing peroxidative damage and inducing the cold response to enhance cold tolerance [8].

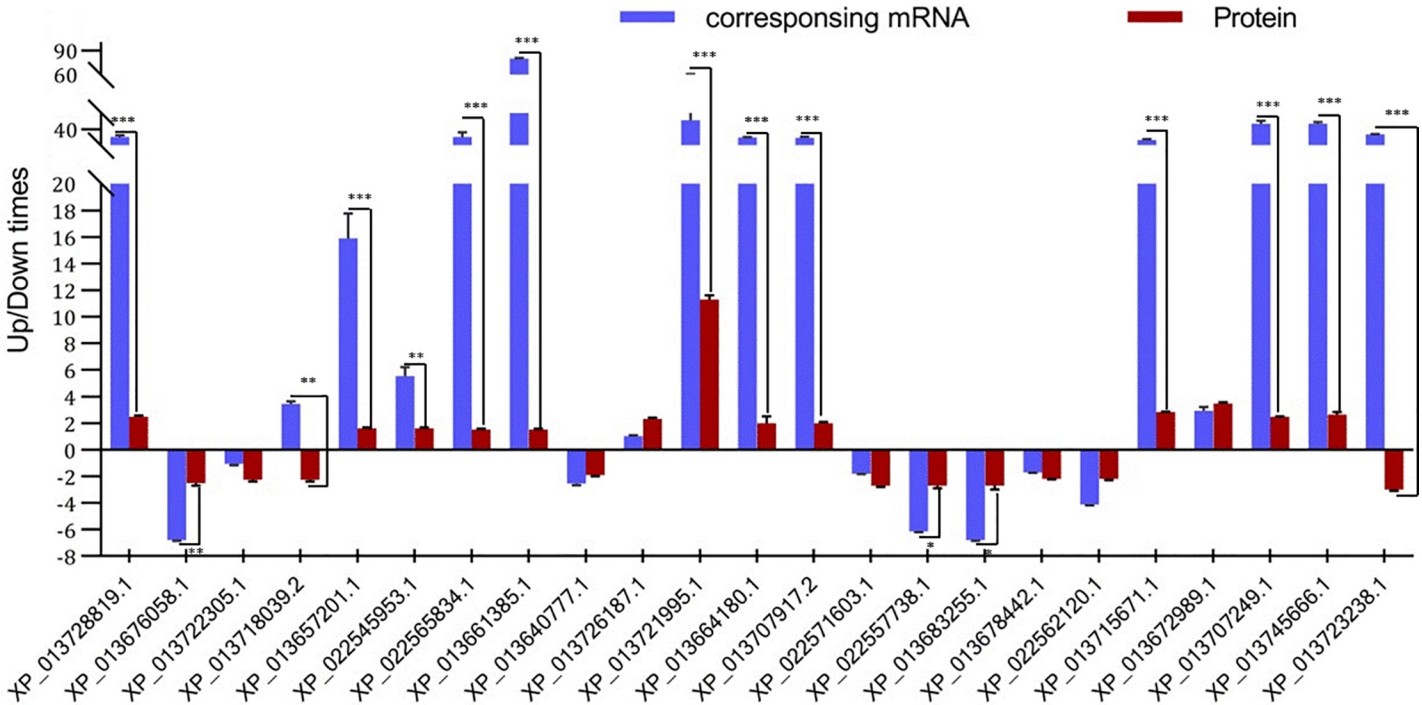

**Fig 6. Transcriptional level of selecting corresponding genes of DAPs between two varieties under cold stress.** The datas were shown as averages of three independent biological replicates±SD.*P<0.05,**P<0.01,***P<0.001.

To prevent ROS from passing over their threshold and causing peroxide damage, ROS must be tightly regulated and maintained at an appropriate level [22]. For this reason, plants in cold regions have evolved antioxidases, including SOD, POD, CAT, and APX, and non-enzymatic scavengers, such as ascorbic acid and glutathione, to avoid the overaccumulation of ROS under cold stress [12]. In this study, the activity of antioxidases SOD, POD, and CAT was higher in the cold-tolerant variety NS than in NF under cold stress. Antioxidases are an induced-enzyme type; thus, their capacity to scavenge ROS can be improved by slightly increasing the availability of ROS as substrates. In contrast, increased antioxidase activity was able to maintain ROS within a suitable range. The higher activity of antioxidases in NS can better regulate ROS level, both avoiding damage caused by the accumulation of ROS and activating cold-signaling pathways to enhance cold tolerance. MDA content and relative conductivity were lower in NS than in NF under cold stress, indicating that the plasma membrane sustained less damaged by ROS peroxidation in NS—and this was most evident during the early stage of cold stress. These results revealed that the response to cold stress was more rapid and intense in NS, potentially explaining its increased cold tolerance.

DIA is a high-throughput analytical method widely used for proteomic analysis [16]. Compared with iTRAQ, DIA is a novel method for proteomic analysis with higher quantitative accuracy and more sensitivity for detecting low-abundant proteins [16]. In this study, DIA proteomic analysis was used to reveal the cold-tolerant molecular mechanism of winter rapeseed at the proteomic level. We have identified proteins with different abundances between the tolerant NS, the sensitive NF, and the control at both 12 h and 24 h of cold treatment. Compared with the cold-sensitive variety NF, 911 unique DAPs were more abundant only under cold stress in NS, including 470 unique DAPs in the T1 treatment (12 h under cold stress) and 437 unique DAPs in the T2 treatment (24 h under cold stress); furthermore, there

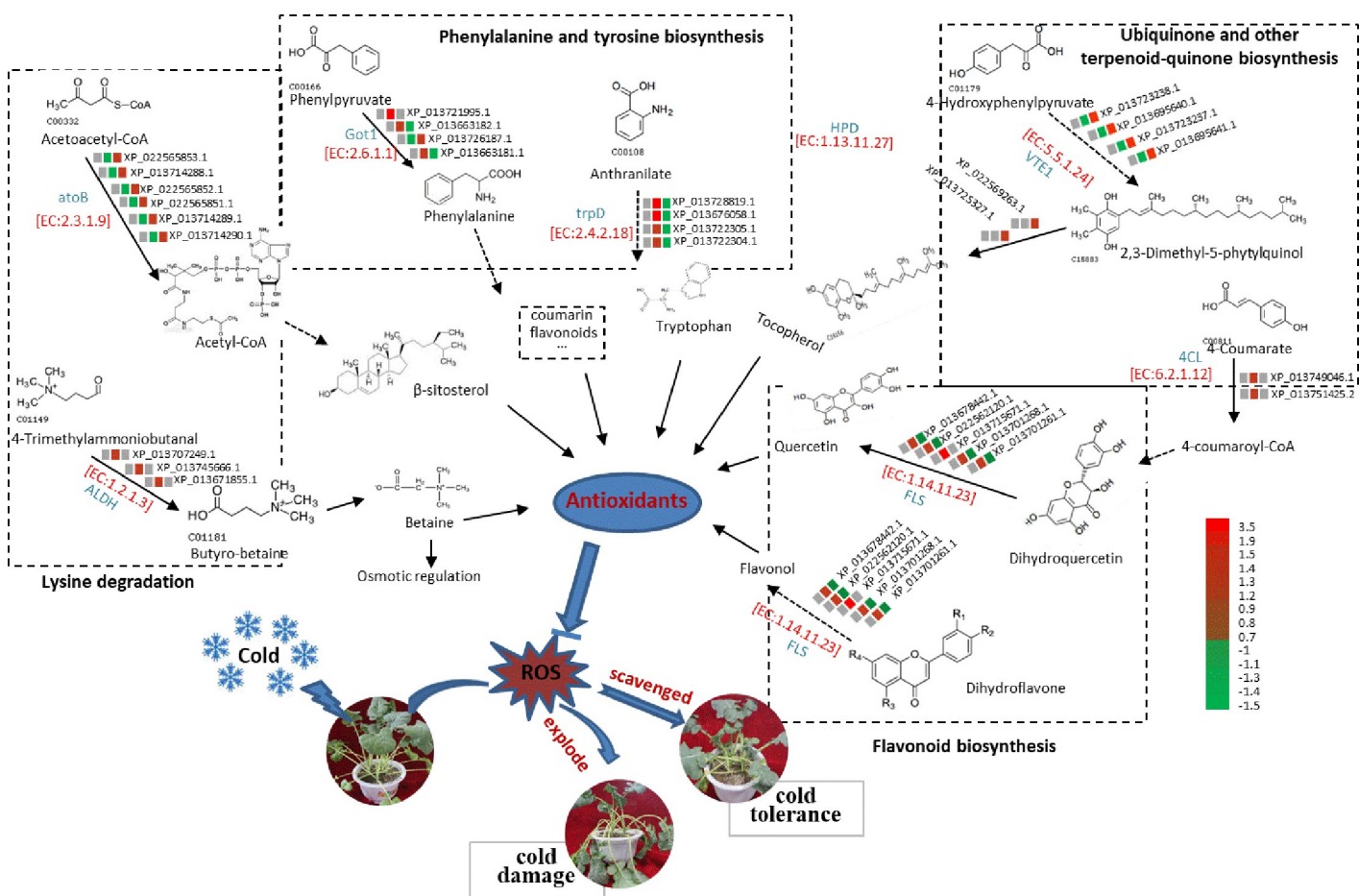

**Fig 7. Differentially abundant proteins (DAPs) related to ROS scavenged under cold stress of winter rapeseed.** Numbers and abbreviations of key enzymes are shown in red and blue text, respectively. Three small squares on a line represented three cold treatments, and color change from green to red meaning DAP accumulative level change from high to low. Arrow with line indicated synthetic direction, line ending with a bar indicated suppressive producing.

were 4 common DAPs in the T1 and T2 treatments. These unique, more highly accumulated DAPs under cold treatment can confer increased cold tolerance to NS. Moreover, more highly abundant DAPs were identified under cold-treatment at 12 h than at 24 h. These results suggest that the early cold response of NS was more rapid and intense.

From a KEGG pathway enrichment analysis, 583 DAPs were significantly enriched in 42 pathways under cold treatment for 12 h, and 721 DAPs were significantly enriched in 30 pathways under cold treatment for 24 h. We have selected four common pathways in the T1 and T2 treatments, and 54 DAPs, including 34 more-abundant DAPs, were annotated in four candidate pathways related to the cold response by affecting the ROS-scavenging process: phenylalanine, tyrosine, and tryptophan biosynthesis (ko00400); flavonoid biosynthesis (ko00941); lysine degradation (ko00310); and ubiquinone and other terpenoid-quinone biosynthesis (ko00130). Some products from the four candidate pathways, such as quercetin [22], flavonoids [23], betaine [24, 25], and tryptophan [10], are important metabolites that act as ROS scavengers via antioxidative reactions in vivo.

Essentially, the phenotypes of plant traits are the result of various metabolic processes. Cold tolerance of plants is a complex quantitative trait, and its formation is associated with intricate metabolic processes [3]. ROS-scavenging metabolism is important for improving cold

tolerance, including phenylalanine metabolism, flavonoid metabolism, and ubiquinone metabolism. Quercetin is one of the most important products of the flavonoid biosynthesis pathway and is a highly active non-enzymatic scavenger of ROS [22, 23, 26]. In this study, the abundance of FLS protein was significantly higher in NS under cold stress; FLS is a synthase catalyzing the biosynthesis of quercetin [27]. Other metabolites, such as flavonol, which is an antioxidant, were produced by catalysis of FLS synthase [28]. Similarly, trpD abundance was distinctly increased in NS under cold stress [29]. TrpD can catalyze the conversion of anthranilate to tryptophan, and tryptophan can enhance the activity of SOD and GSH-PX as well as suppress the accumulation of MDA. The formation of butyro-betaine, the precursor of betaine, via ALDH catalysis is necessary for ensuring a sufficient supply of betaine [30], which is a non-enzymatic ROS scavenger that participates in the cold-tolerant pathway of winter rapeseed. The abundance of SOD and CAT was significantly increased in NS. Therefore, we hypothesize that these proteins may play a crucial role in the ROS-scavenging process related to cold tolerance in winter rapeseed.

## Methods

### Plant materials, cold stress treatment and physiological

The '17NTS57' (abbreviated as "NS," cold-tolerant, greater than 90% survival rate under −26˚C during overwintering) and 'NQF24' (abbreviated as "NF," cold-sensitive, with a 0% overwintering rate) winter rapeseed (*Brassica napus* L.) varieties were used in this study; these varieties were provided by Liu Pro from Gansu Agricultural University (Lanzhou, China). Seedlings were grown in plastic pots until they have six leaves in a climate room at 20˚C under a 14 h light/8 h dark photoperiod. For cold stress treatment, the seedlings were placed into a freezing chamber at -4˚C for 12 h (treatment 1, T1) and 24 h (treatment 2, T2), and without treatment as the control (CK or T0). Subsequently, the leaves of plants were collected, immediately frozen in liquid nitrogen, and stored at -80˚C. Each sample was pooled from three plants, and three biological replicates were conducted. The activity of SOD, POD, and CAT were estimated following the procedures of Zhang et al. [31]. Relative electrolyte leakage (REL) was determined by a digital conductometer DDS11A (Leici Instrument Factory, Shanghai, China) according to Bajji et al [32]. MDA, soluble sugar, and free proline content were measured according to the methods of Yang et al. [33], Buysse et al. [34], and Bates et al. [35], respectively.

### Protein extraction

The total proteins of the leaves from each sample were extracted. Samples were ground to power in liquid nitrogen and then dissolved in 2mL of lysis buffer (8 M urea, 2% SDS, 1×Protease Inhibitor Cocktail (Roche Ltd. Basel, Switzerland)), followed by sonication for 30 min on ice and centrifugation at 13 000 rpm for 30min at 4˚C. The supernatant was then transferred to a fresh tube. For each sample, proteins were precipitated with ice-cold acetone at -20˚C overnight. The precipitations were cleaned with acetone three times and re-dissolved in 8 M urea by sonication on ice. Protein quality was determined using SDS-PAGE.

### Protein digestion

A BCA Protein Assay Kit was used to determine the protein concentration of the supernatant. Fifty μg proteins extracted from leaves were suspended in 50 μL of solution, was concentrated by adding 1ul of 1 M dithiotreitol at 55˚C for 1 h, and was alkylated by adding 5 ul of 20 mM iodoacetamide in the dark at 37˚C for 1 h. The sample was then precipitated using 300 ul of

prechilled acetone at -20˚C overnight. The precipitate was washed twice with cold acetone and was resuspended in 50 mM ammonium bicarbonate. Finally, the proteins were digested with sequence-grade modified trypsin (Promega, Madison, WI) at a substrate/enzyme ratio of 50:1 (w/w) at 37˚C for 16 h.

## High PH reverse phase separation

The peptide mixture was re-dissovled in buffer A (buffer A: 20 mM ammonium formate in water, pH 10.0, adjusted with ammonium hydroxide), and was then fractionated by high pH separation using an Ultimate 3000 system (Thermo Fisher scientific, MA, USA) connected to a reverse-phase column (XBridge C18 column, 4.6mm × 250 mm, 5 μm, (Waters Corporation, MA, USA). High pH separation was performed using a linear gradient from 5% B to 45% B over 40 min (B: 20 mM ammonium formate in 80% ACN, pH 10.0, adjusted with ammonium hydroxide). The column was re-equilibrated at the initial condition for 15 min. The column flow rate was maintained at 1mL/min, and the column temperature was maintained at 30˚C. Ten fractions were collected; each fraction was dried in a vacuum concentrator for the following step.

## DDA: Nano-HPLC-MS/MS analysis

The peptides were re-dissolved in 30 μL of solvent C (A: 0.1% formic acid in water) and analyzed by on-line nanospray LC-MS/MS on an Orbitrap Fusion Lumos (Thermo Fisher Scientific, Bremen, Germany) coupled to a Nano ACQUITY UPLC system (Waters Corporation, Milford, MA). Ten μL of peptide sample was loaded onto the trap column (Thermo Fisher Scientific Acclaim PepMap C18, 100 μm × 2 cm), with a flow of 300 nL/min and subsequently separated on the analytical column (Acclaim PepMap C18, 75 μm × 15 cm) with a set gradient. The column flow rate was maintained at 500 nL/min with a column temperature of 40˚C. An electrospray voltage of 2.1 kV versus the inlet of the mass spectrometer was used.

The mass spectrometer was run under the data dependent acquisition mode and was automatically switched between MS and MS/MS modes. The parameters were as follows: (1) MS: scan range (m/z) = 350–1200; resolution = 120000; AGC target = 500000; maximum injection time = 60 ms; include charge states = 2–6; Filter Dynamic Exclusion: exclusion duration = 30 s; (2) HCD-MS/MS: resolution = 50,000; AGC target = 1000000; maximum injection time = 100 ms; collision energy = 35%; stepped CE = 5%.

## Database search

Raw data of the DDA were processed and analyzed by Spectronaut Pulsar 11.0 (Biognosys AG). Pulsar was set up to search the database of UniProt assuming that trypsin was the digestion enzyme. Carbamidomethylation (C) was the fixed modification, And oxidation (M) was the variable modifications.

## DIA: Nano-HPLC-MS/MS analysis

The peptides were re-dissolved in 30 μL of solvent C (A: 0.1% formic acid in water) and analyzed by on-line nanospray LC-MS/MS on an Orbitrap Fusion Lumos (Thermo Fisher Scientific, Bremen, Germany) coupled to a Nano ACQUITY UPLC system (Waters Corporation, Milford, MA). Ten μL of peptide sample was loaded onto the trap column (Thermo Fisher Scientific Acclaim PepMap C18, 100 μm × 2 cm), with a flow of 300 nL/min and was subsequently separated on the analytical column (Acclaim PepMap C18, 75 μm × 15 cm) with a set gradient.

The column flow rate was maintained at 500 nL/min with a column temperature of 40˚C. An electrospray voltage of 2.1 kV versus the inlet of the mass spectrometer was used.

The mass spectrometer was run under data dependent acquisition mode and was automatically switched between the MS and MS/MS modes. The parameters were as follows: (1) MS: scan range (m/z) = 350–1200; resolution = 120,000; AGC target = 500000; maximum injection time = 60 ms;; (2) HCD-MS/MS: resolution = 50,000; AGC target = 1000000; maximum injection time = 100 ms; collision energy = 35%; stepped CE = 5%.

## Data analysis

Raw data of the DIA were processed and analyzed by Spectronaut Pulsar 11.0 (Biognosys AG) with default parameters (BGS Factory Settings (default)). After conducting a Student's *t*-Test, different expressed proteins were filtered based on the following criteria: Q-value <0.05 and Absolute AVG (Average) log2 ratio>0.58.

## Protein functional annotation and enrichment analysis

Proteins were annotated against the GO, KEGG and COG/KOG database to determine their functions. Significant GO functions and pathways were examined within differentially expressed proteins based on P≤0.05.

## Total RNA extraction and real-time quantitative PCR

Total RNA in leaves was extracted per the manufacturer's instructions (TIANGEN Biotech (Beijing) Co., Ltd.), and RNA integrity was assessed by electrophoresis. The RNA was reverse-transcribed per the manufacturer's instructions (PrimeScript™ RT Reagent Kit with gDNA Eraser, TaKaRa) to obtain single-chain cDNA. Three replicates were performed for each sample. After measuring the concentration, the cDNA was stored at -20˚C. The primers used in real-time quantitative PCR are listed in S5 Table. The Actin gene (Accession ID: 106427432) was used as an internal reference, and the relative expression of each gene was analyzed by the $2^{-\Delta\Delta Ct}$ method [36].

## Concluding remarks

In this study, DIA-based proteomic technology was used to investigate the differential proteomics of two winter rapeseed cultivars (NS and NF of *B. napus*) at -4˚C for 12 h (T1), 24 h (T2), and at room temperature (T0). A total of 1978 DAPs in NST0_NFT0, 1235 DAPs in NST1_NFT1, and 1543 DAPs in NST2_NFT2 were identified. We identified a total of 34 shared DAPs that were abundant in NST1_NFT1 and NST2_NFT2 but not in NST0_NFT0. Based on the functional analysis, these 34 proteins may improve the cold tolerance of winter rapeseed by regulating metabolic pathways related to ROS scavenging (namely, phenylalanine, tyrosine, and tryptophan biosynthesis; flavonoid biosynthesis; lysine degradation; and ubiquinone and other terpenoid-quinone biosynthesis). Several candidate proteins (FLS, TrpD, ALDH, 4CL, GOT1, atoB, CA4H, and VTE1) related to antioxidants or scavengers may scavenge ROS to improve the cold tolerance of winter rapeseed. In addition, more proteins were more highly accumulated in NS in T1 but not in T2. Thus, the cold response appears to be more rapid and intense in NS than in NF, which might explain why NS is more cold resistant than NF.

## Supporting information

**S1 Table. Sequencing result data.**
(XLSX)

**S2 Table. DAPs in NST0_NFT0, NST1_NFT1, and NST2_NFT2.**
(XLSX)

**S3 Table. Gene ontology (GO) analysis of differentially expressed cold-related genes of NS and NF.**
(XLSX)

**S4 Table. Candidate pathways and candidate proteins information.**
(XLSX)

**S5 Table. Primer of qRT-PCR.**
(XLSX)

**S6 Table. Candidate proteins information.**
(XLSX)

## Acknowledgments

**Disclaimer:** There are no copyright disputes.

## Author Contributions

**Data curation:** Wenbo Mi, Xiaoyun Dong, Chunmei Xu.

**Investigation:** Wenbo Mi, Jiaojiao Jin, Ya Zou, Mingxia Xu, Guoqiang Zheng, Xiaodong Cao, Xinling Fang, Caixia Zhao.

**Software:** Wenbo Mi, Chao Mi.

**Writing – original draft:** Wenbo Mi.

**Writing – review & editing:** Wenbo Mi, Zigang Liu.

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
