## [Decision Letter · Decision Letter 0]

21 Sep 2020

PONE-D-20-23256

Comparative proteomics analysis reveals the molecular mechanism of enhanced cold tolerance through ROS scavenging in winter rapeseed (Brassica napus L.)

PLOS ONE

Dear Dr. Liu,

Thank you for submitting your manuscript to PLOS ONE. After careful consideration, we feel that it has merit but does not fully meet PLOS ONE’s publication criteria as it currently stands. Therefore, we invite you to submit a revised version of the manuscript that addresses the points raised during the review process.

We look forward to receiving your revised manuscript.

Kind regards,

Maoteng Li

Academic Editor

PLOS ONE

Journal Requirements:

2.Thank you for stating the following in the Acknowledgments Section of your manuscript:

[This work was supported by the National Natural Science

Foundation of China (31660404), National Key Basic Research and Development

Program (2018YFD0100500), University of Gansu Province Scientific Research

Achievement Transformation and Cultivation Project (2018D-13), Special Fund for

the Construction of Modern Agricultural Industrial Technology System of Gansu

Province (17ZD2NA016-4).]

 [The funders had no role in study design, data collection and analysis, decision to publish, or preparation of the manuscript.]

5. Please include your tables as part of your main manuscript and remove the individual files. Please note that supplementary tables (should remain/ be uploaded) as separate "supporting information" files

6. Please ensure that you refer to Figure 7 in your text as, if accepted, production will need this reference to link the reader to the figure.

7. Please upload a new copy of Figure 3 and 4 as the detail is not clear. Please follow the link for more information: https://blogs.plos.org/plos/2019/06/looking-good-tips-for-creating-your-plos-figures-graphics/" https://blogs.plos.org/plos/2019/06/looking-good-tips-for-creating-your-plos-figures-graphics/

Reviewers' comments:

Reviewer's Responses to Questions

**Comments to the Author**

1. Is the manuscript technically sound, and do the data support the conclusions?

Reviewer #1: Yes

Reviewer #2: Yes

2. Has the statistical analysis been performed appropriately and rigorously? 

Reviewer #1: Yes

Reviewer #2: No

3. Have the authors made all data underlying the findings in their manuscript fully available?

Reviewer #1: Yes

Reviewer #2: Yes

4. Is the manuscript presented in an intelligible fashion and written in standard English?

Reviewer #1: No

Reviewer #2: No

5. Review Comments to the Author

Reviewer #1: In this manuscript, the authors reported two winter rapeseed cultivars, cold tolerant and cold sensitive, were used to reveal the morphological, physiological, and proteomic characteristics in leaves of plants after treatment at different temperature points. The candidate pathways of DAPs involved were analyzed and ROS scavenging pathway may play function in rapeseed cold tolerance. The paper is poorly displayed and following items should be modified.

1. The format of references is incorrect, please correct it.

2. The description of Result 2.2, “…identified based on the criteria of p < 0.005 for the adjusted p-value…”, is confused.

3. An obvious descriptive error existed in Methods 4.1, “…store at -80 reaching concentrations thaEach sample….”.

4. Abstract part, the first line, it is should be “cultivars” not “culvers”. It is recommended to find native English speaker to modify the language of the paper.

4. It is confused to understand if the proteomics performed with biological repetition?

Reviewer #2: In this work, the authors performed comparative proteomics analysis reveals the molecular mechanism of enhanced cold tolerance through ROS scavenging in winter rapeseed (Brassica napus L.). Although this work is interesting, the manuscript is not well organized, and the data are not well presented. Therefore, the paper needs to be thoroughly revised.

1. The authors must carefully proof-read the manuscript to minimize typographical, grammatical, and bibliographic errors. Some typographical errors, some nonsensical or confusing sentences should be corrected.

2. In Abstract, “Two winter rapeseed culvers, “NF” (cold tolerant) and “NS” (cold sensitive), were used to reveal the morphological, physiological, and proteomic characteristics……”. What does ‘culvers’ mean? ‘culvers’ or ‘cultivars’? Please check the spelling.

3. In Abstract, authors refer to two winter rapeseed culvers “NF” and “NS”, among, “NF” represent cold-tolerant cultivar and “NS” represent cold-sensitive cultivar. However, in Introduction and Methods parts, authors refer to “strong cold-tolerant variety “NS” and '17NTS57' (abbreviated as “NS”, cold-tolerant…) and 'NQF24' (abbreviated as “NF”, cold-sensitive…) winter rapeseed (Brassica napus L.). These seem to be particularly controversial. Please confirm and explain why?

4. In Plant materials, cold stress treatment and physiological part, authors say “the cold-tolerant variety ‘17NTS57’ beyond 90% survival rate under −26 °C of extreme low temperature during overwintering, please provide data supporting the result or cite the related references.

5. In results part, ‘(figure2B )’ should put it after the sentence ‘By comparing all of the DAPs between the two varieties at three treated time points, the Venn diagram shows the number of more- or less-abundant DAPs as well as specific and commonly abundant DAPs in the different groups.’

6. In general, the technical approaches of RT-qPCR seem fine. However, statistics for figure 6 are not given. Also, statistics for DAGs (real-time quantitative RT-PCR) abundance changes (eg p-values) are not given (not in method and not in legend). What method were used? ANOVA? One or two way? What was the level of significance?

7. Authors selected 23 DAPs for detecting the mRNA levels using qRT-PCR. However, the protein names are not given in figure 6 or in Result part. The protein names are replaced by Numbers 1, 2, 3, 4, 5……

8. In Result part, no any information and sentences related to figure 7. Please add …..!

9. The results lack systematic in-depth analyses. Authors think cold-tolerant rapeseed enhance cold tolerance through ROS scavenging, but no related candidate proteins were in-depth analyzed and studied.

10. Figure quality is too low and the legends need to be reorganized and rephrased.

6. PLOS authors have the option to publish the peer review history of their article (what does this mean?). If published, this will include your full peer review and any attached files.

Reviewer #1: No

Reviewer #2: No

---

## [Author Response · Author response to Decision Letter 0]

16 Oct 2020

We have carefully revised and checked the manuscript in accordance with the review comments to make the paper more complete and meet the requirements of the journal.

---

## [Decision Letter · Decision Letter 1]

10 Nov 2020

PONE-D-20-23256R1

Comparative proteomics analysis reveals the molecular mechanism of enhanced cold tolerance through ROS scavenging in winter rapeseed (Brassica napus L.)

PLOS ONE

Dear Dr. Liu,

Thank you for submitting your manuscript to PLOS ONE. After careful consideration, we feel that it has merit but does not fully meet PLOS ONE’s publication criteria as it currently stands. Therefore, we invite you to submit a revised version of the manuscript that addresses the points raised during the review process.

We look forward to receiving your revised manuscript.

Kind regards,

Maoteng Li

Academic Editor

PLOS ONE

Reviewers' comments:

Reviewer's Responses to Questions

**Comments to the Author**

1. If the authors have adequately addressed your comments raised in a previous round of review and you feel that this manuscript is now acceptable for publication, you may indicate that here to bypass the “Comments to the Author” section, enter your conflict of interest statement in the “Confidential to Editor” section, and submit your "Accept" recommendation.

Reviewer #1: All comments have been addressed

2. Is the manuscript technically sound, and do the data support the conclusions?

Reviewer #1: Yes

3. Has the statistical analysis been performed appropriately and rigorously? 

Reviewer #1: Yes

4. Have the authors made all data underlying the findings in their manuscript fully available?

Reviewer #1: Yes

5. Is the manuscript presented in an intelligible fashion and written in standard English?

Reviewer #1: Yes

6. Review Comments to the Author

Reviewer #1: The authors modified all the concerns we raised, I think it can be accepted with a minor revision.

1. Please write the full name for the abbreviation of AVG when it first appeared.

2. In methods part, the real-time quantitative PCR, “The Actin gene was used as an internal reference…..”, the accession number for the actin gene used should be listed in the paper.

7. PLOS authors have the option to publish the peer review history of their article (what does this mean?). If published, this will include your full peer review and any attached files.

Reviewer #1: No

---

## [Author Response · Author response to Decision Letter 1]

11 Nov 2020

First of all, and thanks the scholars who reviewed this manuscript for their valuable comments on this paper, thanks the responsible editor of the manuscript for your letter, as well as the important guiding significance to our researches. We have carefully revised and checked the manuscript in accordance with the review comments to make the paper more complete and meet the requirements of the journal.

---

## [Editor Report · Decision Letter 2]

19 Nov 2020

Comparative proteomics analysis reveals the molecular mechanism of enhanced cold tolerance through ROS scavenging in winter rapeseed (Brassica napus L.)

PONE-D-20-23256R2

Dear Dr. Liu,

We’re pleased to inform you that your manuscript has been judged scientifically suitable for publication and will be formally accepted for publication once it meets all outstanding technical requirements.

Kind regards,

Maoteng Li

Academic Editor

PLOS ONE
---

## [Editor Report · Acceptance letter]

23 Dec 2020

PONE-D-20-23256R2 

**Comparative Proteomics Analysis Reveals the Molecular Mechanism of Enhanced Cold Tolerance Through ROS Scavenging in Winter Rapeseed (*Brassica napus* L.)**

Dear Dr. Liu:

I'm pleased to inform you that your manuscript has been deemed suitable for publication in PLOS ONE. Congratulations! Your manuscript is now with our production department. 

Kind regards, 

on behalf of

Dr. Maoteng Li 

Academic Editor

PLOS ONE